# Paleopathological Changes in Animal Bones from Croatian Archaeological Sites from Prehistory to New Modern Period

**DOI:** 10.3390/vetsci10050361

**Published:** 2023-05-18

**Authors:** Tajana Trbojević Vukičević, Kim Korpes, Martina Đuras, Zoran Vrbanac, Ana Javor, Magdalena Kolenc

**Affiliations:** 1Department of Anatomy, Histology and Embryology, Faculty of Veterinary Medicine, University of Zagreb, 10000 Zagreb, Croatia; tajana@vef.unizg.hr (T.T.V.); martina.duras@vef.unizg.hr (M.Đ.); mkolenc@vef.unizg.hr (M.K.); 2Department of Radiology and Physical Therapy, Faculty of Veterinary Medicine, University of Zagreb, 10000 Zagreb, Croatia; zvrbanac@vef.unizg.hr (Z.V.); ajavor@vef.unizg.hr (A.J.)

**Keywords:** paleopathology, domestic animals, Croatia, macrostructural changes, radiologicalexamination

## Abstract

**Simple Summary:**

With the goal of contributing to the correct diagnosis of pathological changes found in archaeozoological material, we give a detailed description of those changes. All the changes were photographed and radiographed. In total, 50 animal remains with altered macrostructure were identified in archaeozoological material excavated from 2010 to 2022 at eight archaeological sites in Croatia.

**Abstract:**

A special part of archaeology, so-called archaeozoopathology or veterinary paleopathology is dedicated to studies of paleopathological changes in animal remains and contributes to the knowledge of ancient veterinary medicine and the history of diseases. In our study, we analyze paleopathological changes determined by gross observation and diagnostic imaging in the animal material originating from eight archaeological sites in Croatia. A standard archaeozoological analysis was carried out and specimens with visually detected macrostructural changes were radiographed. In total, 50 animal remains with altered macrostructure were identified in the archaeozoological material excavated from 2010 to 2022 at eight archaeological sites in Croatia. According to the taxonomic analysis, most of the bones with macrostructural changes originated from cattle (N = 27, 54% of the total number of bones with macrostructural changes), followed by the bones of small ruminants (N = 12, 24%) and pigs (N = 8, 16%). The horse, carnivore and chicken were represented with one bone each (2%). Radiological examination showed that three samples (6%) had a regular bone macrostructure, i.e., no pathological changes were visible upon radiological examination. The majority (64%) of pathologically altered bones are a consequence of keeping/working, followed by traumatic causes (20%). Changes in the oral cavity were found in 10% of specimens. Our study showed that gross examination will continue to be the primary method for the identification of pathologically altered remains in archaeozoological material. However, diagnostic imaging techniques such as radiography should be implemented to confirm or exclude suspected alterations and to help the classification of the specimen by etiology.

## 1. Introduction

Animal remains (bones, teeth and horns) recovered from archaeological sites represent an important material that should be analyzed to gain a complete picture of the social and economic status of an ancient society. Data on animals that lived in the proximity to humans—including species, number, sex and age classes of these animals, estimated body size and mass—are crucial in the analysis of ancient human–animal relationships, i.e., domestication, husbandry, breeding, animal production and rituals. However, the fact that not all remains originate from healthy animals must be taken into consideration during data interpretation. Moreover, structural changes found in archaeozoological material may indicate diseases, breeding and husbandry failures or working overload.

A special part of archaeology, so-called veterinary paleopathology [1] or archaeozoopathology [2] is dedicated to studies of paleopathological changes in animal remains and contributes to the knowledge about ancient veterinary medicine and the history of diseases [2,3,4,5]. There are several challenges in the study of palaeopathology on animal remains when compared to human remains. For example, human bones are most frequently preserved as the whole skeleton of an individual, whereas animal bones are often excavated as disarticulated skeletons found in waste pits [3,6]. Furthermore, the effect of deposition and the age of domestic animals, which is related to both their use by people and the postmortem preservation of skeletal parts, should be considered during the analysis of suspected paleopathological changes. Moreover, diseases themselves may alter the bone structure and consequently affect the preservation status of the animal remains. Different infectious diseases, inflammations, neoplasia, degenerative changes and trauma can lead to macro and micro changes in bone structure. The changes can be visible in the form of bone tissue hypertrophy or hypotrophy/osteolysis. Those deviations from the standard bone shape that are usually observed macroscopically can indicate any of the pathologies of the skeletal system. Various types of damage to the bone may be interpreted alternatively as having taken place in vivo or postmortem [7].

Methods of choice in palaeopathology research are gross examination and diagnostic imaging, i.e., radiography and computed tomography [8]. These methods have proven useful to determine changes in the standard bony macrostructure. However, pathological changes are frequently not pathognomonic, so the cause of those changes often remains unknown. In many cases, only the general etiology of the pathological change can be reached [1]. Furthermore, to archaeozoologists, only pathological changes in the skeletal system, teeth and horny structures are available, which represent a small range of animal diseases in general [9]. As a result, a lower prevalence of diseases may be estimated for a certain population [10].

Classification of paleopathological changes determined on animal remains may be performed according to the geographical location, the chronological period of the archaeological site, etiological factors, animal species or skeletal region [2]. Unfavorable husbandry practices have been determined as a common cause of paleopathological changes [6]. Most often, these types of changes can be found in working animals, i.e., cattle and horses [11,12]. In the horse, changes have been observed on the thoracic and lumbar vertebrae and on the distal limb bones. These changes have been associated with intensive long-term riding and loading [2,5,13,14,15,16]. However, such changes have been found in almost complete skeletons of horses of warrior nations, which were often buried with their riders [17,18]. In phalanges and the calcanei of cattle originating from Viking Age and Middle Age deposits in Sweden, three types of pathological changes have been found: exostoses, lipping and depressions/lesions [19]. It is presumed that these changes are caused by using cattle for pulling heavy loads. A low percentage (less than 15%) of cattle from England sites had severe changes in the phalanges and metapodials, which indicated that just a small number of cattle were constantly used for draught [20]. On the other hand, Lin et al. [21] report on the usage of cattle for traction in China that took place from earlier periods (Erlitou and Early Shang Dynasty) to later periods (Late Shang Dynasty). The bones of cattle from their study did not show severe pathological changes like one can find in modern traction animals; moreover, the incidence of pathologies present in modern meat cattle was low. The authors concluded that most of the cattle were used for light work and traction.

Marković et al. [15] described changes in limb bones related to hard-working animals based on ruminants and equine remains found at two medieval sites in Serbia. Most pathological changes were found on the metacarpal and metatarsal bones and phalanges in the form of exostoses and osteophytes with a prevalence of 0.85%. Moreover, teeth and jaws showed pathological changes that were related to poor diet. Animal material originating from Estonian archaeological sites showed numerous dental anomalies and, to a lesser extent, bone deformities due to fractures or joint inflammation [22]. Siegel [9] gave an overview of paleopathological findings in animal remains originating from 18 archaeological sites in Britain dating from the Neolithic to medieval times. Many pathological changes were found at the level of the oral cavity (33%), followed by arthropathies (20%) and fractures (10%). Even in domestic fowl, paleopathological changes have been observed in the form of shaft osteopetrosis in long bones [23].

Paleopathological changes expressed as structural changes in the bones, teeth or horns can contribute to the age determination of the animal since some diseases occur more frequently in young animals while others are exclusively diseases of older-age specimens. A low occurrence of diseases can indicate a young animal population, whereas a high incidence of diseases may be due to an older animal population, inbreeding or weak husbandry [9].

Data on pathological changes found in animal bones, teeth and horns from archaeological sites in Croatia are rare, despite intensive archaeozoological research that has been ongoing for decades in that region. In our study, we analyzed paleopathological changes determined by gross observation and diagnostic imaging in the animal material originating from eight archaeological sites. Those archaeological sites are in different geographical regions of Croatia, i.e., Eastern and Western Continental Croatia and Northern Dalmatia, and have been populated during different time periods. For each observed paleopathological change, we give a detailed description followed by presentations on photographs and radiographs. With this work, we would like to contribute to the correct diagnosis of pathological changes observed in archaeozoological material, which should prevent misleading interpretations and prove helpful in the analysis of human–animal relationships in ancient societies.

## 2. Materials and Methods

### 2.1. Archaeological Sites

This study was performed as part of the complete archaeozoological analysis of eight archaeological sites from Croatia. Five medieval archaeological sites (depending on the site, from the 14th to the end of the 16th century), one Late Bronze Age (late 15th–14th century BC) till Roman Period (till the end of the 6th century) site, one Late Bronze/Iron Age (late 15th–14th century BC/9th–5th century BC) and one church which existed from the Older Iron Age (9th–5th BC) to the Early Modern Period (17th–19th century) have been included in this study. The locations of the sites are in Eastern and Western Continental Croatia and Northern Dalmatia (Figure 1). All archaeological data included in this study, i.e., the dating and purpose of the site, the period of origin of the archaeozoological material and the time of excavation, have been determined and published previously by archaeologists responsible for the excavations of a certain site. Excavations have been performed by the Croatian Conservation Institute, Department of Archaeology, University of Zadar and International Center for Underwater Archaeology in Zadar with the permissions and under the supervision of the competent Croatian Ministry.

#### 2.1.1. Eastern Continental Croatia

Gorjani (GOR) is the archaeological site in Osijek-Baranja County which consists of the medieval urban settlement with fortifications built in the second half of the 14th century till the first half of the 15th century [24]. The excavation of the archaeozoological material took place in 2015 in and around the chapel Sveta Tri Kralja located within the settlement. It is estimated that the animal material originated from the middle of the 16th century until the end of the 17th century, i.e., from the Early Modern Period.

Osijek Ciglana–Zeleno polje (OCZP) is in Osijek-Baranja County. Here, the earliest settlement was established during the Middle Bronze Age and was continually settled till the early phase of the Late Bronze Age. After a long hiatus, the site was settled again in the late phase of the Early Iron Age and in the Late La Tène period [25]. The animal material was excavated between 2015 and 2016 and originated from the early (15th century BC) and late (late 15th–14th century BC) phases of the Middle Bronze Age.

Bijela is in Bjelovar-Bilogora County and consists of remains of the Benedictine monastery that was built on elevated ground above a stream surrounded by a defensive wall. It was one of Slavonia’s most important Benedictine centers in the 14th and 15th centuries. The church and remains of a monastery with a sacristy and a chapter house were excavated in 2012 when animal remains dated from the end of the 19th century, i.e., from the Early Modern Period, were found [26].

#### 2.1.2. Western Continental Croatia

Plemićki grad Vrbovec (PGV) is in Zagreb County and includes remains of a Romanesque burg with a perimeter wall, palace, defensive tower, yard with a cistern and wooden economic buildings [27,28]. The excavation of animal remains took place in 2010, 2014, 2018 and 2019, and it is estimated that the material originated from the Late Middle Ages (13th century, end of the 13th and beginning of the 14th century) and the Early Modern Period (15th and 16th centuries). 

Stari grad Barilović (BAR) is a medieval city in Karlovac County. It was populated in three phases: a feudal fortress (15th and 16th century), a military fortress (17th and 18th century) and a city under civil administration [29]. Animal material was excavated in 2018 and originated from the 15th and 16th centuries, i.e., from the High to Late Middle Ages. 

Stari grad Krčingrad (KRČ) is in the Plitvice Lakes National Park in Lika-Senj County and consists of a triangular defense tower surrounded by a defensive wall made of travertine. The tower was built at the end of the 13th or the beginning of the 14th century (High/Late Middle Ages) [30]. Animal bones were excavated from 2020 to 2022 and dated to the Late Middle Ages.

#### 2.1.3. Northern Dalmatia

Nadin–Gradina (NADIN) is in Zadar County and was settled by indigenous Liburnian culture from the Late Bronze Age (11th century BC). In the 1st century BC, Nadin was incorporated into the Roman Empire. At the end of the Roman Period (6th century AC), the site was abandoned but occupied again during the 15th–17th century by Ottomans and Venetians [31,32]. Animal material was excavated in 2015, 2016, 2019 and 2020 and it originated from the Older Iron Age (9th–5th century BC), Antiquity and/or Late Antiquity (end of the 1st/2nd till end of 6th century) and Late Middle Ages/Early Modern Period (15th/16th century).

St Nicholas’ Church (ST NIC) is in Zadar County immediately adjacent to the shore. Here, remains of a church built in the 11th century were found; in the 14th century, another church was erected at the same site, and a third church was built after 1760 in baroque style at the site of the earlier church. The church was desacralized in the year 1798 and converted into a military barrack, later into a military hospital [33]. Animal remains were excavated in 2014 and originated from the 9th–5th century BC (Older Iron Age) and from the end of the 18th century (Early Modern Period).

### 2.2. Archaeozoological Analysis and Diagnostic Imaging

At the archaeological sites, animal remains, i.e., bones, teeth and horns, were excavated, labeled, stored and transported to the Department of Anatomy, Histology and Embryology at the Faculty of Veterinary Medicine, University of Zagreb, Zagreb, Croatia. To remove the soil remains, the animal material was washed with tap water and then dried at room temperature. After drying, a standard archaeozoological analysis was carried out, which included skeletal and taxonomic identification [34,35,36,37], age and gender estimation [34,36,37], and notification of taphonomic and pathological changes [38,39,40]. 

In the cases when macrostructural changes were detected visually with the naked eye and a magnifying lamp, the animal remains were separated for further analysis. First, the observed macroscopical changes were described and photographed. Furthermore, all bones with macrostructural changes were radiographed at the Department of Radiology, Ultrasound Diagnostics and Physical Therapy, Faculty of Veterinary Medicine, University of Zagreb. For radiography, a Siemens Multix Compact K + LG flat panel digital detector (55 kV and 8 mAs) was used. Each bone was radiographed in three views: lateral, sagittal and orthogonal. Radiographic scans were used to confirm or discard macroscopically observed alterations from a standard bone structure. To analyze the possible cause of the pathological changes and draw conclusions about their etiology, the findings were classified according to the radiological diagnosis into four groups that have been modified according to Thrall [41]: (1) metabolic changes, (2) inflammatory and/or neoplastic changes, (3) degenerative changes, (4) traumatic changes. Furthermore, the obtained radiological diagnoses were grouped according to von den Driesch [42] and Bartosiewicz [2] as follows: (1) oral cavity pathology: dental anomaly, metabolic, inflammatory/neoplastic, degenerative changes; (2) keeping/working disorders: metabolic, inflammatory/neoplastic, degenerative changes; (3) trauma.

## 3. Results

In total, 50 animal remains with altered macrostructure were identified in the archaeozoological material excavated from 2010 to 2022 at eight archaeological sites in Croatia. Table 1 shows basic data for each animal remain, e.g., sample number, archaeological site and period of origin, animal species, etc.

All animal remains were determined as complete bones or bone fragments, in the text further referred to as bones. No macroscopical alterations were found on horns or isolated teeth. The prevalence of bones with altered macrostructure in the archaeozoological material per site is shown in Figure 2. According to the taxonomic analysis, most of the bones with macrostructural changes originated from cattle (N = 27, 54% of the total number of bones with macrostructural changes, N = 50), followed by the bones of small ruminants (N = 12, 24%) and pigs (N = 8, 16%), while the horse and chicken were represented with one bone each (2%). The carnivore sample (P23) referred to four fused lumbar vertebrae that were counted as one remain (2%). The taxonomic distribution of the bones expressed per archaeological sites showed no significant pattern related to the site or to the period of origin (Table 1 and Table 2).

Each bone with an altered structure that was observed during the gross examination was macromorphologically described and radiologically examined. In three (P12, P13 and P21) out of fifty bones that were macromorphologically assessed as having visible alterations, radiological examination showed a regular bone macrostructure, i.e., no pathological changes were visible upon radiological examination.

### 3.1. Skull

Pathological changes were found on eight mandibles (16% of 50 bones with macrostructural changes). No other bone of the skull showed structural alterations. The eight mandibles originated from the archaeological sites Plemićki grad Vrbovec (PGV), Stari grad Krčingrad (KRČ), Bijela, Osijek Ciglana–Zeleno polje (OCZP) and Nadin–Gradina (NADIN). According to the NISP analysis, they belonged to different specimens, i.e., three cattle, three small ruminants and two pigs. The left bovine mandible P7 showed a loss of the corticalis with uneven edges at the level of the head of the mandible (caput mandibulae). The size of the corticalis loss was 2 × 0.5 cm and the underlying spongious bone tissue was clearly visible. The lesion was located on the medial half of the head of the mandible, and it reached the rostral edge of the articular surface. Radiological examination displayed signs of reduction in the cortical and trabecular bone, which is consistent with osteoporosis (Figure 3).

Similar macrostructural changes were found in the left bovine mandible P10. However, the indentation defect extended to the caudal edge of the articular surface and divided the surface into two parts, a smaller medial one and a larger lateral one. The edges of the defect were smooth. In the mandible P10, radiological examination confirmed the corticalis defect with secondary arthritic changes (Figure 4).

The third left bovine mandible (P49) showed signs of irregular, rough bony swelling with sponge-like appearance at the level of the mandibular body (corpus mandibulae). Radiological examination showed irregular periosteal proliferations compatible with signs of osteomyelitis, but these changes were also similar to sunburst-like periosteal proliferations that are consistent with osteosarcoma (Figure 5).

The right small ruminant mandible P15 showed a thickening of the mandibular body, 2.5 cm in length, just ventrally to the premolars. The swelling reached dorsally the alveolar margin. Radiological examination showed hyperostosis of the spongious bone tissue, whereas the compact bone tissue was regular in terms of width. Rostrally and laterally to the premolar alveolus, two periosteal reactions with smooth margins were visible. The left goat mandible (P18) showed signs of bone loss with a sponge-like appearance. The change was found rostrally to the mental foramen on the lateral surface of the mandibular body. Radiological examination showed signs of osteomyelitis with a periosteal reaction. Additionally, a third non-erupted premolar (P3) was seen deep in the alveolus, just under the deciduous P3. The third small ruminant mandible (P37), left-sided, showed thickening of the buccal surface of the mandibular body ventral to teeth alveoli (from the level of the third premolar, P3, to the second molar, M2). The fourth premolar (P4) and first and second molars (M1, M2) were missing because of taphonomic factors. Radiological examination confirmed signs of osteomyelitis (Figure 6).

In the right pig mandible (P21), receding alveolar margins were visible on both buccal and lingual sides around the right first molar (M1); moreover, the occlusal surface of M1 was more worn, and at least half of the tooth root was exposed. However, radiological examination showed a regular macrostructure. In a second pig mandible (P22), left-sided, a thickening of the mandibular body ventral to the premolar teeth was seen. The thickening affected both the lingual and buccal surfaces and stretched from the second premolar (P2) to the M1. The radiological examination displayed signs of fracture and reparation, i.e., formation of the callus (Figure 7).

### 3.2. Ribs and Vertebrae

Pathological changes in ribs were represented by only one sample (1 out of 50 bones, 2%), whereas three samples revealed altered vertebrae (3 out of 50 bones, 6%). The rib and one axis were from a small ruminant, one lumbar vertebra was from a pig, and four fused canine lumbar vertebrae were treated as one sample. The rib and vertebrae originated from the archaeological sites Osijek Ciglana–Zeleno polje (OCZP), Plemićki grad Vrbovec (PGV) and Nadin–Gradina (NADIN). The small ruminant rib (P9) showed bone swelling in the middle of the body. Radiological examination showed an old fracture (status post fracturam) with the loss of the cortical bone tissue and enthesophytes in the middle of the defect. The small ruminant axis (P47) showed loss of bone tissue at the level of the dens. The dens appeared small, rounded and irregular in shape. Radiological examination showed subchondral cysts in the axis body located close to the dens surrounded by mild periosteal and periarticular reactions compatible with secondary degenerative arthritic changes. The lumbar vertebra of the pig (P11) showed bone tissue swelling on the ventral and caudal part of the vertebral body with bone tissue loss on the caudal extremity with sponge-like appearance. Radiological examination displayed ankylosis of the ventral part of the body and periosteal bone proliferation of the transverse process. The only carnivore sample with observed macrostructural changes (P23), most likely a dog, revealed four fused lumbar vertebrae, presumably L2–L5. The vertebrae were completely fused along the ventral side of their bodies with distinct bony swellings. Radiological examination confirmed ankylosis of the ventral part of the vertebral bodies. 

### 3.3. Thoracic Limb Bones

The thoracic limb bones with macrostructural changes represented 34% of all affected bones (N = 17 out of 50 affected bones). Most frequently, macrostructural changes were observed on metacarpal bones (N = 9; 18%), followed by the humerus and radius, which were represented by three bones each (6% of the sample per bone). The ulna and carpal bones were represented by one sample each (2% of the sample per bone). The thoracic limb bones originated from the archaeological sites Plemićki grad Vrbovec (PGV), Stari grad Krčingrad (KRČ), Stari grad Barilović (BAR), Gorjani (GOR), St. Nicolas’ Church (ST NIC) and Nadin–Gradina (NADIN). According to the NISP analysis, they belonged to different animal species. The description of the macrostructurally altered bones is given in the order according to their position in the thoracic limb, from proximally to distally. The distal part of the humerus of a small ruminant (P8) showed cranially and medially newly formed bone tissue on the distal metaphysis. Radiologically, it was determined as enthesophyte on the craniomedial side of the distal metaphysis. The pig humerus (P16) showed swelling cranially at the lateral part of the condyle which was radiologically determined as subchondral sclerosis of the compact bone tissue. The only avian bone in the sample was a humerus of the chicken (P36) with a pronounced periosteal thickening of the diaphysis through its whole length. Radiologically, a formation of the callus within status post fracturam was found (Figure 8).

In total, three radii (6%) were affected by pathological changes, two from small ruminants and one from a horse. One radius of a small ruminant (P29) showed bone swelling of the diaphysis which was approximately three times wider than usual. Radiological examination showed status post fracturam with the formation of the callus in the middle and the distal third of the diaphysis (Figure 9).

Another small ruminant radius originated from a sheep (P50) and showed a small bone swelling just distal to the proximal articular surface. Radiological examination showed enthesophyte on the proximal metaphysis. In the whole sample, only one equine radius (P24) was found with macrostructural changes. A prominent ridge was found on the distal and medial side of the diaphysis in the form of bone swelling. Radiologically, two linear periosteal growths were determined at the same position, 1.6 cm and 3.3 cm in length, respectively. The only ulna in the sample originated from a small ruminant (P20). It was bent caudally, which was confirmed by radiological examination and described as a slight degree of ulna deviation. A radial carpal bone (os carpi radiale) of the cattle (P48) showed a noticeable thickening on the dorsal surface. Radiologically, osteolysis with bone tissue swelling was determined. Metacarpal bones with altered macrostructure belonged to six cattle (P1, P19, P25, P28, P33, P43), one sheep (P45), one goat (P46) and one pig (P3). In three bovine metacarpals (P25, P28, P33), only the distal half of the bone was preserved. In all three bones, the same change was observed in the form of inequality of articular surfaces of the head, i.e., the medial side (os metacarpale III) of the articular surface of the head was wider than the lateral side. However, radiological findings differed for these three macromorphologically similar changes. Osteosclerosis and enthesophyte at the wider medial articular surface of the head were observed in P25. Metacarpal P28 showed a bone cyst laterally in the wider articular surface of the head. The metacarpal P33 showed a mild degree of asymmetry of the lateral and medial articular surface of the head, both macroscopically and radiologically. The bovine metacarpal bone marked as P1 was well preserved and missed only the distal part. It was slightly bent and thickened along the diaphysis, whereas bone loss on the proximal epiphysis was observed. Radiological examination showed the thickening of the compact bone tissue of the diaphysis with the reduced width of the medullary cavity. In two bovine metacarpals (P19 and P43), proximally on the palmar side, just ventral to the articular surface, bone swellings were found. Additionally, in P43 the fifth metacarpal was fused to the fourth. Radiologically, P19 showed a linear swelling of the caudolateral side of the periosteum, 2.7 cm in length. In P43, status post fracturam was determined with the thickening of the compact bone tissue at the level of the proximal metaphysis. The right fourth metacarpal bone of a pig (P3) had visible swelling in the middle of the body on the palmar side. Radiologically, irregular margins of the compact tissue were observed and status post fracturam suspecta was diagnosed. On the lateral side of the proximal epiphysis of the sheep metacarpal bone (P45), the formation of a relatively large bone swelling/newly formed bone was visible. Radiologically, it corresponded to a 5.4 cm long enthesophyte. The proximal epiphysis of the metacarpal bone of the goat (P46) carried a bone swelling/newly formed bone almost axially on the proximal articular surface. The presence of the osteophyte was confirmed radiologically. 

### 3.4. Pelvic Limb Bones

The pelvic limb bones with macrostructural changes appear in our sample to a lesser extent when compared to thoracic limb bones. In total, nine bones of the pelvic limb with macrostructural changes were determined, which makes up 18% of the whole sample (N = 50). Most frequently, macrostructural changes were observed on tarsal bones (N = 4, 8% of the sample), and two metatarsal bones were affected (4%), whereas the hip bone, femur and tibia were represented by one sample each (2% of the sample per bone). The pelvic limb bones originated from the archaeological sites Plemićki grad Vrbovec (PGV), Krčingrad (KRČ) and Nadin–Gradina (NADIN). According to the NISP analysis, they belonged to different species, i.e., five cattle, three pigs and one small ruminant. The description of the macrostructurally altered bones is given in the order according to their position in the pelvic limb, from proximally to distally. The acetabulum of a bovine hip bone (P14) showed bone swelling around the acetabular notch and on the lunate surface. Radiological examination confirmed signs of arthrosis. 

The distal epiphysis of a pig’s femur (P4) showed bone loss in the form of an irregular depression on the lateral condyle. It was radiologically visualized as two round smaller lucent areas with mild sclerosis at their margins. Such a periosteal defect with slightly sclerotic margins refers to status post fracturam suspecta. The tibia of a pig (P17) showed bone swelling in the distal part of the diaphysis. That part of the bone appeared three times wider than in a regular bone. Fibula was fused to the tibia all along its length. Radiological examination showed a status post fracturam with the formation of the callus. On two bovine calcanei (P12, P13), macromorphologically, a slight thickening was observed on the plantar surface just distal to the calcaneal tuber. However, the radiological findings showed a regular bone macrostructure, i.e., no pathological changes were found. In the talus of a pig (P2), the loss of bone tissue was observed along the dorsal and the plantar surfaces as round to oval depressions with smooth edges. Radiological examination showed signs of osteoporosis and arthrosis. In one bovine centroquartal tarsal bone (P35) on the proximal articular surface, a big depression (1.5 × 0.5 cm) was found, which was radiologically described as osteolysis. The distal half of a bovine metatarsal bone (P6) showed an unequal width of the articular surfaces of the head, with the lateral articular surface (os metacarpale IV) being wider. Radiological examination showed osteosclerosis of the wider, lateral head (Figure 10). Another metatarsal bone belonged to a small ruminant (P44). A thickening was found in the middle of the diaphysis. Radiological examination showed status post fracturam with the formation of the callus.

### 3.5. Bones of the Metapodium and Acropodium

In a certain number of bones with pathological changes (N = 12, 24% of the sample), the skeletal position and body side could not be determined. This was the case in one (2%) unfused distal epiphysis of a bovine metapodium and eleven (22%) bovine phalanges. These bones originated from the archaeological sites Plemićki grad Vrbovec (PGV), Gorjani (GOR), St. Nicolas’ Church (ST NIC) and Nadin–Gradina (NADIN). In the unfused distal epiphysis of a bovine metapodium (P5), a small irregular crescent-shaped indentation was detected on the lateral articular surface of the head (os metacarpale/metatarsale IV). Radiological examination showed early degenerative changes most likely caused by OCD (osteochondrosis dissecans). In nine bovine proximal phalanges (P26, P30, P31, P32, P38, P39, P40, P41, P42), swellings of the compact bone tissue were found all along the bone. Proximal phalanges P39 (Figure 11) and P41 were macroscopically very similar to the others, but radiologically, they showed osteolysis of the distal epiphysis and bone swelling.

The proximal phalanx P26 was macroscopically wider than usual, especially at the level of the diaphysis. Radiologically, the periosteum showed a small area of interruption of the continuity with micro lucent areas. Two bovine middle phalanges (P27 and P34) showed bone swellings. In P34, it occupied the whole medial side of the diaphysis. Radiological examination showed a suspected status post fracturam with periosteal bone swelling with flat margins. Secondary arthritic changes and subchondral sclerosis were also found (Figure 12). 

Bone swelling in phalanx P27 was much smaller and located on the distal and lateral sides of the diaphysis. Radiologically, bone swelling axially on the proximal epiphysis was observed with the formation of the enthesophytes on the distal and lateral sides of the diaphysis. Table 3 shows the classification of all specimens according to etiological (oral cavity, keeping/working, trauma) and radiological classification (metabolic, inflammation/neoplasm, degenerative). Given that we did not find any dental anomalies in our research, they are not listed in Table 3.

## 4. Discussion

Our comprehensive archaeozoological analyses of material originating from eight different Croatian archaeological sites and different periods included 50 bones of domestic animals, which were macromorphologically assessed to be pathologically altered. The incidence of pathologically affected bones per archaeological site was extremely low; in most of the involved sites, it was below 1%, except Gorjani, which showed a slightly higher incidence (1.41%). Likewise, our study showed that the number of identified specimens (NISP) by archaeological site and the incidence of pathologically altered bones were not related, e.g., Barilović was one of the sites with the highest number of animal remains NISP (4737); however, it was one of the sites with the lowest number of pathologically altered bones (N = 2, 0.19%). Such a low percentage of pathologically altered bones in the entire archaeozoological material from a site was encountered in other research; moreover, Bartosiewicz [2] stated that more than ten pathologically altered bones may be expected to occur only in assemblages that contain over ten thousand (!) identifiable bone specimens. Furthermore, during the calculations of the prevalence of diseases in ancient animals, one should take into account several known facts about archaeozoological material; on the contrary, a lower prevalence of diseases may be calculated in a certain population [10]. Namely, an archaeozoological material includes only horn/antler structures, teeth and (mainly) disarticulated individual animal bones [9], whereas other organic systems and tissues are not preserved and pathological changes in these structures remain unknown. Moreover, animal remains are frequently found in waste pits within the settlement or at some other collection point in a populated building (house, castle, church). Rarely do the remains originate from the same individual [3,6]. Furthermore, faunal assemblages may have experienced multiple deposition events [1].

In our study, all pathologically changed bones were subjected to radiological examination, which proved to be helpful in the identification of pathological alterations in archaeozoological material. On the other hand, suspected pathological alterations may also be excluded with the help of radiological examination. For example, during the gross examination in our study, two bovine calcanei and one pig mandible showed suspected changes which were not proved during the radiological examination, i.e., radiography of these three bones showed regular bone macrostructures. 

Both calcanei (P12, P13) originated from the site Plemićki grad Vrbovec (PGV) from the 16th century, i.e., from the Early Modern Period. Those bones belonged to cattle older than 3.5 years from urban settlements. In such settlements, the cattle were kept for secondary products, primarily milk [28]. Visible structural changes were found proximally on the plantar side of the bone, which is the place of insertion of the plantar ligament (ligamentum plantare longum), a very strong and the most important plantar ligament of the tarsus [37]. We presume that the visible changes represented pronounced attachment points of the ligament, which is typical for older individuals. However, we estimate that the animal was not old enough to eventually develop degenerative changes, such as enthesophytes, that would already be outlined radiologically. 

The third bone with suspected pathological alterations that were not proved radiologically was a pig’s right mandible (P21) from the OCZP site which originated from the late phase of the Middle Bronze Age (the late 15th–14th century BC). That mandible showed a sunken alveolar margin at the area of the right first molar (M1), on both the buccal and lingual surface. The occlusal surface of M1 was more worn and at least half of the tooth root was exposed. Radiologically, a regular bone macrostructure was found. Regardless of the radiological findings, we presume that a pathological process led to osteolysis of the bone tissue in that area. The lesion, which was very likely the result of a periodontal disease, healed spontaneously and there were no alterations traceable on the radiological examination. An almost identical change was recorded on the sheep mandible from Portchester Castle with a visible alveolar bone recession as the result of periodontal disease (Figure 7b in [40]). This inflammatory disease in animals can, among other things, be caused by incorrectly worn or erupted teeth. In the case of ancient jaws, the evidence for periodontal disease is to be seen in receding alveolar margins, localized periostitis, pockets between teeth and abnormally loose teeth [40]. 

Considering paleopathological classification according to von den Driesh [42] and Bartosiewicz [2], most of the pathological changes found in our study may be linked to the working/keeping etiology, which was determined in 32 bones (64%); traumatic changes were found in 10 bones (20%) and oral cavity diseases were observed in 5 bones (10%).

### 4.1. Working/Keeping Etiology

Most of the pathological changes determined in our sample, i.e., 32 (64%) bones, were classified into the keeping/working etiology group. Within this group, the presumed diseases that caused the pathological changes in bones visible by gross and/or radiological examinations might be due to metabolic disorders, inflammation/neoplasm or degenerative processes. Radiologically, we determined that only one specimen had pathological changes that should be linked to a metabolic cause (ulna P20 of a small ruminant). No similar cases have been described in the published literature; therefore, we did not speculate about the disease that led to the metabolic disorder resulting in the structural alteration of the ulna (P20). In four bovine bones, we concluded that the structural changes were caused by inflammation/neoplasm. Most of the pathological changes that were classified in the working/keeping etiology group were caused by degenerative processes and were represented in 27 bones (54% of the whole sample). All alterations caused by inflammation/neoplasm were found in bovine bones which originated from the Nadine site: one tarsal centroquartale bone (P35) and two proximal phalanges (P39 and P41) were from the Antiquity period, while the carpal radial bone was from the Early Iron Age. Stevanović et al. [43] stated that osteolytic changes in the form of cavitation and sinuses, like those visible in P35, were most probably signs of suppurative processes, i.e., abscesses, within the bone tissue. Degradation of the articular surface with the periosteal reaction and the formation of marginal osteophytes seen in P39, P41 and P48 resembled the articular infection described by Stevanović et al. [43].

In the working/keeping etiology group, most alterations were linked to degenerative changes (N = 27), most frequently observed in bovine bones (N = 17 out of 27), followed by small ruminants (N = 5), pigs (N = 3) and one horse and one canine sample. Similarly, cattle were the most affected species (N = 10) in the sample studied by Marković et al. [15], who described only one altered bone originating from a small ruminant. Furthermore, the most frequently affected bovine bones were metacarpal bones and proximal phalanges, which is also like the findings of Marković et al. [15], who described most of the changes at the distal bones of the limbs, i.e., metacarpals and metatarsals. 

Asymmetry of the lateral or the medial articular surface of the bovine metapodial head, thickening of the compact bone tissue of the metapodial diaphysis, swellings of the compact bone tissue, in the form of exostoses and enthesophyte near the joint articular surface, are among the most common paleopathological degenerative changes on metacarpals from our sample. Similar pathological changes were described as grooving on the articular surfaces, eburnation, an extension of the articular surfaces and peripheral exostoses by Baker and Brothwell [40] and Thomas [44]. Moreover, osteophytes and enthesophyte were determined by Marković et al. [15]. 

Bovine phalanges were the most numerous bones with pathological changes represented in our sample (N = 11). We found nine bovine proximal and two bovine medial phalanges with bone swellings mostly at the level of the diaphysis. In two proximal phalanges (P39 and P41), the pathological process affected the distal articular surface, which was diagnosed upon radiological examination as osteolysis. Thomas [44] described osteophyte formation, eburnation and grooving of the proximal and/or distal articular surface in four bovine phalanges, two proximal and two media, and stated that the pathological changes observed in their sample were caused by osteoarthritis. Moreover, Thomas [44] presumed that the most important intrinsic factor for the pathological changes found at the level of the articular surfaces was age. Etiologically, primary and secondary types of osteoarthritis can be distinguished. In primary osteoarthritis, in which there is no external predisposing factor, the age of the animal and reduced blood flow that leads to ischemic necrosis of the joint cartilage and consequent degenerative changes can be taken into consideration. Secondary osteoarthritis may develop as a result of development diseases of the joints or some exogenous predisposing factors, such as trauma and physical burdening [45]. Morphologically, changes in the joints are the same, starting with cartilage erosion on the articular surface where is the highest loading, after which ulceration and visible fibrillation follow. Expansion of the pathological process to subchondral bone led to sclerosing changes and eburnation, sometimes with subchondral formation of cysts [46]. The whole process ends with marginal osteophytes (exostoses) and with the extension of the articular surface [43,45].

However, osteoarthritis does not explain the swellings found at the level of the diaphysis in the phalanges from our sample. In our opinion, bone swellings at the level of the diaphysis might be explained as enthesophytes, i.e., abnormal bone growth at the site of ligament or tendon attachment. Enthesophytes usually develop because of a reparation process taking place at the lesions in tendons or ligaments [47]. We presume that severe damage to metacarpophalangeal and proximal interphalangeal joints and their ligaments resulted in bone swellings at the level of the phalangeal diaphysis observed in our sample.

Bartosiewicz et al. [11] compared the changes on articular surfaces of bovine bone remains and modern Romanian draft cattle, and due to the marked similarity of the pathomorphological changes, they concluded that such changes are the best indicators of cattle exploitation for work. Another study from Bartosiewicz [48] observed that pathological alteration occurred more often on bovine metapodial and phalanges from the Roman period and Middle Ages than in prehistoric cattle with an explanation of the increased use of animals, especially cattle, in later historical periods. Unfortunately, regardless of the different periods, the number of bone samples of cattle in our study was too small for such comparisons.

While degenerative changes on the distal parts of the legs in cattle and horses are generally recognized in the literature as a pathology of "working animals" [11,49], in small ruminants, the cause is most likely old age, and thus breeding and keeping due to secondary exploitation, especially milking in goats and wool production in sheep [2]. 

Pathological alterations along the spine have been described in many paleopathological studies and several authors offer explanations of certain conditions in different animal species. Degenerative changes in the spine in horses are most often associated with riding [5,13,14,15,16,18,50] and in cattle with draft use [9,15,20,21,22,51]. In other species, such changes are mainly associated with age and are the result of one of the three main ways: natural aging, repetitive strain syndrome (RSI) and longevity in animals exploited for secondary products [7]. In our sample, we observed unusual degenerative changes on the axis of a small ruminant originating from the Early Iron Age from the Nadin site. We presume that the changes were the result of RSI because of accumulated repeated trauma. 

Although pigs are single-meat-purpose animals, ankylosis diagnosed in our sample on one lumbar vertebra of a pig was most likely related to disproportionately large body weight [2]. However, the observed change could also possibly be related to the older age of the individual, because we were not able to determine if the sample originated from a pig or a wild boar. Ankylosis or spondylosis deformans ancylopoetica is a degenerative disease characterized by a fusion of the vertebrae and occurs in all domestic animals but is most common in older dogs [45]. The ankylosis of our carnivore sample is manifested on four lumbar vertebrae completely fused along the ventral side with distinct bony swellings, i.e., osteophytes. The etiology of this degenerative disease is multicausal; however, it is most often associated with traumatic lesions of the nucleus pulposus [43]. Moreover, age, gender and body size may also contribute to this pathological change. Based on the hypothesis that the spondylosis deformans is an indicator for the use of ancient dogs, e.g., if they were used to pull or carry loads, Latham and Losey [52] systematically analyzed 136 modern non-transport dogs, 19 sled dogs, and 241 wolves from North America and Europe for the occurrence of this spine pathology. The results indicated that this pathology is not a reliable skeletal indicator of dogs used in transport because the disease is prevalent in both dogs and wolves, regardless of their occupational histories.

### 4.2. Traumatic Changes

According to the number of affected specimens, traumatic changes formed the second-largest etiological group. Pathological changes that were determined by gross examination and radiologically as traumatic lesions were detected in 10 bones (20% of the total sample, N = 50). 

Changes in eight bones were diagnosed as status post fracturam; that is, the changes occurred as a result of a healed bone fracture, which is manifested by the formation of a callus at the fracture site. Such changes were observed in three bones of small ruminants, three bones of a pig, one bovine bone and one bone of a chicken. Most frequently, pathological changes related to trauma were found in the metapodials (P3, P43, P44), whereas the rib (P9), humerus (36), radius (P29), tibia (P17) and mandible (P22) were represented by one bone each. Fractures are one of the most common pathological changes in the archaeozoological assemblage, and they are the easiest to recognize and diagnose, although the causes of the trauma are rarely evident [2,40,53]. The causes of fractures can be divided into three major groups: acute as a consequence of direct or indirect trauma, stress or fatigue fractures caused by repetitive stress, and pathological fractures caused by congenital, metabolic or infectious process which disables the bone to withstand normal biomechanical stress [53].

Bartosiewicz [2] recorded a broken Chalcolithic goat radius healed with a light dislocation and a large callus at the Horum Höyük site. Additionally, he reported on a sheep metatarsus with a large callus across the diaphysis from the Shahr-I Sokhta site and stated that bones of the autopodium tend to be more exposed to external trauma than more proximally segments of the zeugo or stylopodium.

In the Estonian archaeological site Parnu from the 14th–15th century, a well-healed goat metacarpal and sheep metatarsal were found [22]. The sheep metatarsal healed without remarkable displacement of the broken ends, which may suggest the use of a splint. However, healed metacarpals have been observed in wild animals and human intervention may not be a factor in this case. 

Broken ribs [22,53] and healed fractured tibias in the pig [2] may be hypothetically related to the tethering of the animals. Teegen and Wussow [54] observed rib and vertebral fractures in pigs and sheep from the 19th and early 20th centuries, suggesting the kicking and beating of animals by humans. However, Teegen [55] argued that such fractures may be from kicking by larger animals. We speculate that the traumatic injuries of mandibles observed in our study are the result of goring in the same herd (small ruminants) or with larger animals.

Descriptions of healed bone fractures in large ungulates (cattle and horses) are rare. It is explained by the fact that the prognosis of leg fractures is usually bad in large ungulates, because the substantial body weight tends to hinder or even prevent good recovery [2]. Moreover, fractures of long bones heal poorly, which led to the slaughtering of animals with fractures and the lack of evidence [53].

Moreover, in the traumatic injuries group, the pig’s distal femur (P4) with periosteal changes and the small ruminant mandible (P15) with periosteal reaction were included. We presume that these changes occurred post-traumatically, but not necessarily because of the bone fracture. Namely, the periosteal reaction can be the result of inflammation of the periosteum caused by trauma or infection. Baker and Brothwell [40] mentioned that localized soft tissue wound infection extending to the bone sometimes leaves characteristic patterns of bone change.

Most of the paleopathological changes were easy to classify into an etiological group by gross examination and/or radiologically. On the other hand, certain specimens were extremely difficult to classify in only one group, i.e., overlapping inevitably occurs. That was true for the bovine middle phalanx (P34), where radiological examination showed a suspected status post fracturam with secondary arthritic changes and subchondral sclerosis, which is a degenerative change.

### 4.3. Oral Cavity Group

In two bovine mandibles that originated from the PGV site, P7 from the Early Modern Period and P10 from the High Middle Ages, a similar pathological alteration was observed. Their caput mandibulae showed an indentation defect on the articular surface, but the radiological diagnosis was different: in P7, osteoporosis (metabolic) was determined, and in P10, degenerative arthritis changes were observed. Metabolic diseases of the joints, as well as necrosis, fractures and septic arthritis, can cause secondary osteoarthrosis [45], i.e., degenerative changes. This example showed that one should be careful establishing a diagnosis based only on macroscopic findings.

The third bovine mandible (P49) originated from the site of Nadin from the Early Iron Age. The radiological diagnosis was osteomyelitis with periosteal reaction, but osteosarcoma may also be taken into account. Osteomyelitis was also diagnosed in a goat (P18) from the Bijela site (Early Modern Period) and a small ruminant (P37) from the Nadin site (Late Middle Ages/Early Modern Periods). Osteomyelitis is defined as an infectious inflammation of bone tissue, usually caused by bacteria in animals, including facultative pathogenic bacteria found in the oral cavity. Osteomyelitis of the mandible is caused by the bacteria Actinomyces bovis (Lumpy Jaw) or periodontal disease [47]. Other causes may be overgrazing [2], poor grazing land, insufficient nutrients during growth and varying amounts of abrasive material in foodstuff [40].

In our research, oral cavity diseases account for only 10% of the total pathological alterations. This is different from changes found in animal bones from five medieval Estonian archaeological sites [22], where the most pathology was related to the oral cavity, and moreover to oligodontia. Likewise, at 18 sites in Great Britain ranging from Neolithic to medieval times [9], oral cavity diseases account for as much as 33%.

## 5. Conclusions

Our study represents the first comprehensive analysis of paleopathological material originating from different archaeological sites in Croatia. The low incidence of animal remains with pathological alterations was similar to archaeological sites in other geographical regions and periods. This is in accordance with the general presumption that pathologically altered bones are difficult to spot in archaeozoological material. The main reason for this may be because animal bones most frequently appear as disarticulated skeleton parts. Moreover, pathological disorders may change bone structures and accelerate their decomposition. Limb bones were the most frequent findings in our sample. In general, phalanges were the most numerous bones with pathological changes which may be due to their compact structure that saved them from major taphonomic influences. Our study showed that gross examination will continue to be the primary method for the identification of pathologically altered remains in archaeozoological material. However, diagnostic imaging techniques such as radiological examination should be implemented to confirm or exclude suspected alterations and to help the classification of the specimen by etiology.

## Figures and Tables

**Figure 1 vetsci-10-00361-f001:**
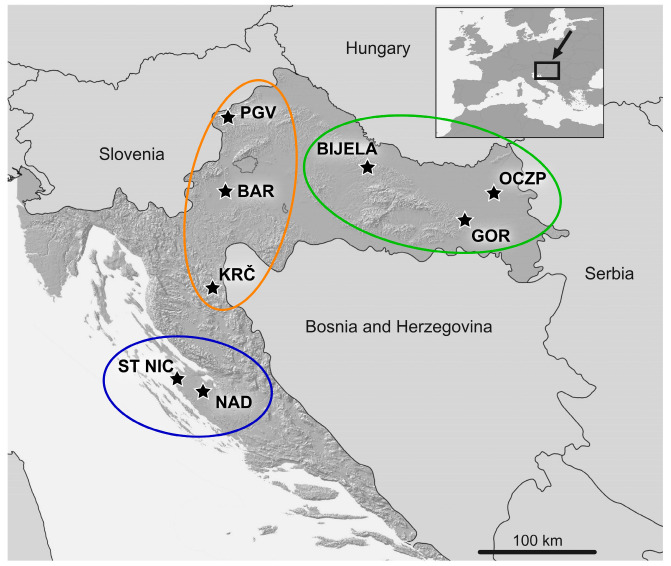
A map of Croatia showing the position of eight archaeological sites, divided into three groups. Orange circle—Western Continental Croatia; green circle—Eastern Continental Croatia; blue circle—Northern Dalmatia. PGV—Plemićki grad Vrbovec; BAR—Barilović; KRČ—Stari grad Krčingrad; BIJELA - Bijela; GOR—Gorjani; OCZP—Osijek Ciglana–Zeleno polje; ST NIC—St Nicholas’ Church; NAD—Nadin–Gradina.

**Figure 2 vetsci-10-00361-f002:**
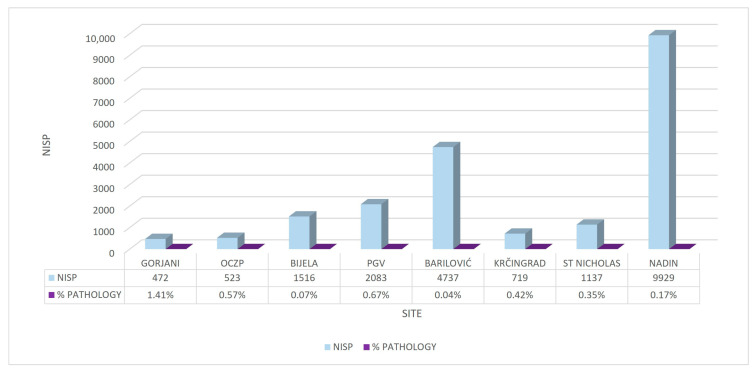
Comparative presentation of the NISP (number of identified specimens) and the percentage share of pathologically altered bones per archaeological site. OCZP—Osijek Ciglana-Zeleno Polje; PGV—Plemićki grad Vrbovec; BARILOVIĆ—Stari grad Barilović; KRČINGRAD—Stari grad Krčingrad; ST NICHOLAS—St. Nicholas Church; NADIN—Nadin-Gradina.

**Figure 3 vetsci-10-00361-f003:**
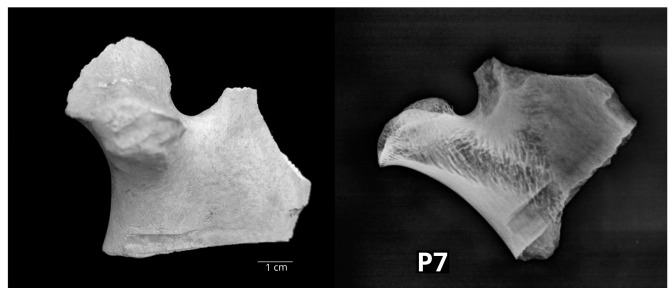
The left bovine caput mandibulae (P7) showed reduced cortical and trabecular bone tissue, which is typical for osteoporosis (**left**—lateral macromorphological view; **right**—lateral radiological projection).

**Figure 4 vetsci-10-00361-f004:**
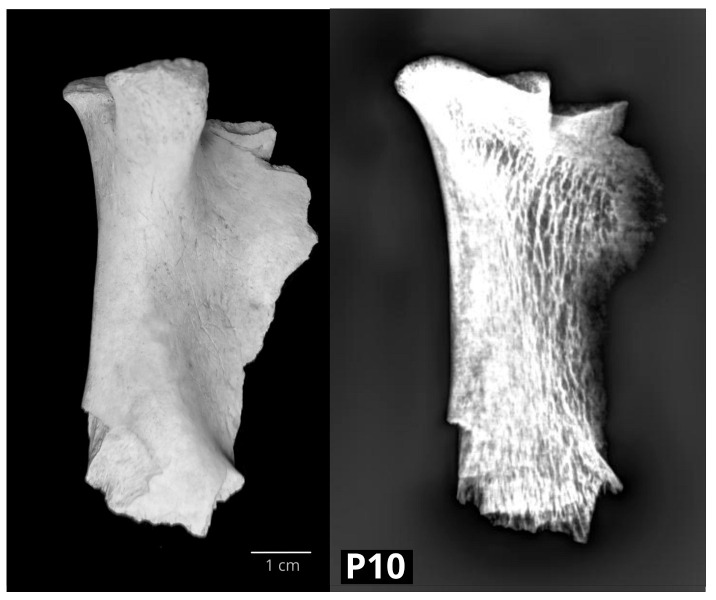
The left bovine caput mandibulae (P10) with a corticalis defect on the articular surface and secondary arthritic changes. Loss of cortical and trabecular bone tissue, typical for osteoporosis, is visible radiologically (**left**—medial macromorphological view; **right**—lateral radiological projection).

**Figure 5 vetsci-10-00361-f005:**
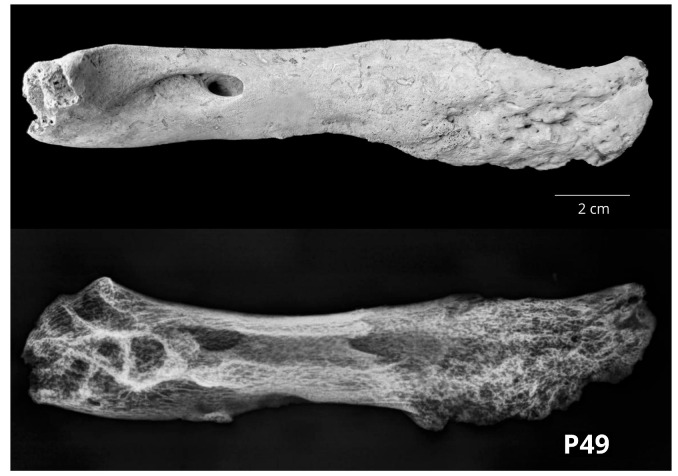
Pars incisiva and margo alveolaris of the left bovine mandible (P49) showing sponge-like swelling typical for periosteal proliferations compatible with osteomyelitis and/or osteosarcoma (**above**—lateral macromorphological view; **below**—lateral radiological projection).

**Figure 6 vetsci-10-00361-f006:**
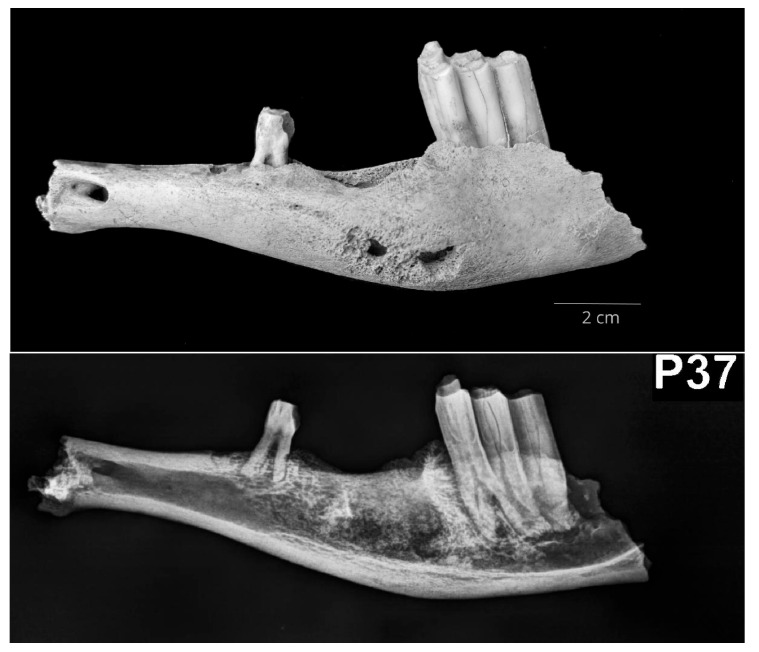
The left small ruminant mandibular body (P37) shows a thickening typical for osteomyelitis at the level of the buccal surface (**above**—lateral macromorphological view; **below**—lateral radiological projection).

**Figure 7 vetsci-10-00361-f007:**
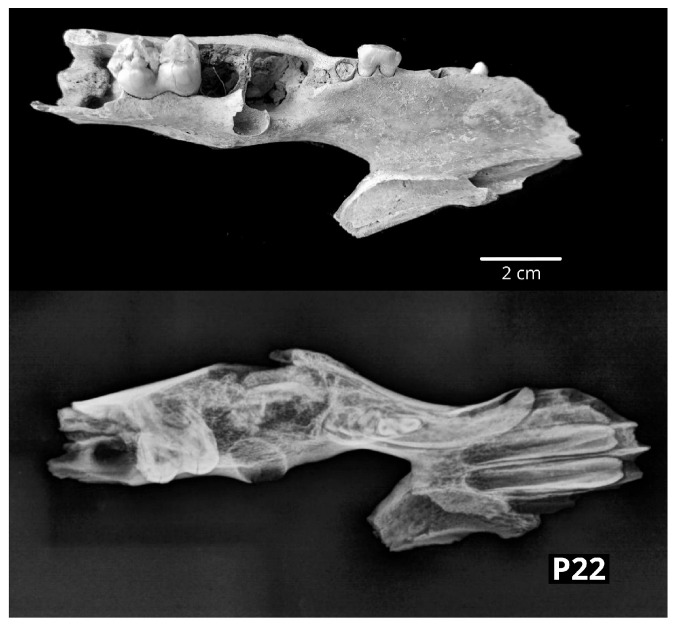
The left mandibular body of the pig (P22) showing a thickening radiologically confirmed as fracture and reparation (**above**—dorsal macromorphological view; **below**—sagittal radiological projection).

**Figure 8 vetsci-10-00361-f008:**
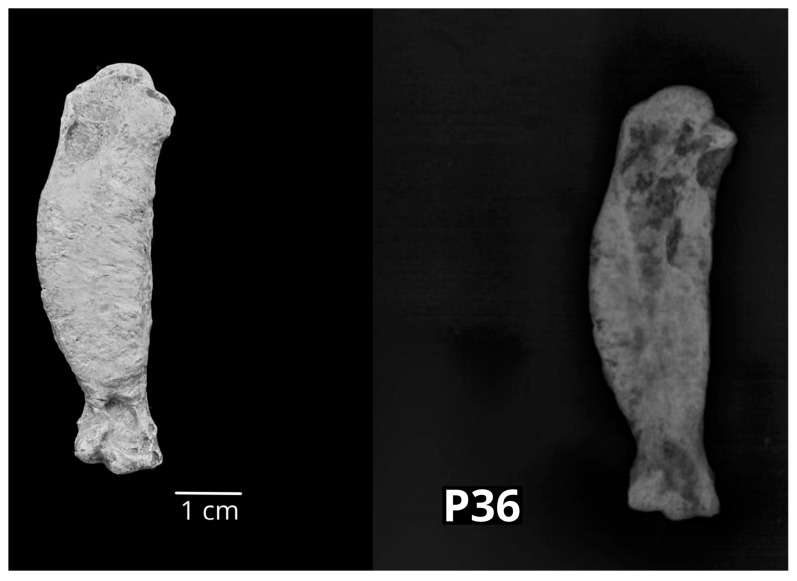
The right chicken humerus (P36) with periosteal thickening of the diaphysis through its whole length. Radiologically, it showed a formation of the callus within status post fracturam (**left**—cranial macromorphological view; **right**—sagittal radiological projection).

**Figure 9 vetsci-10-00361-f009:**
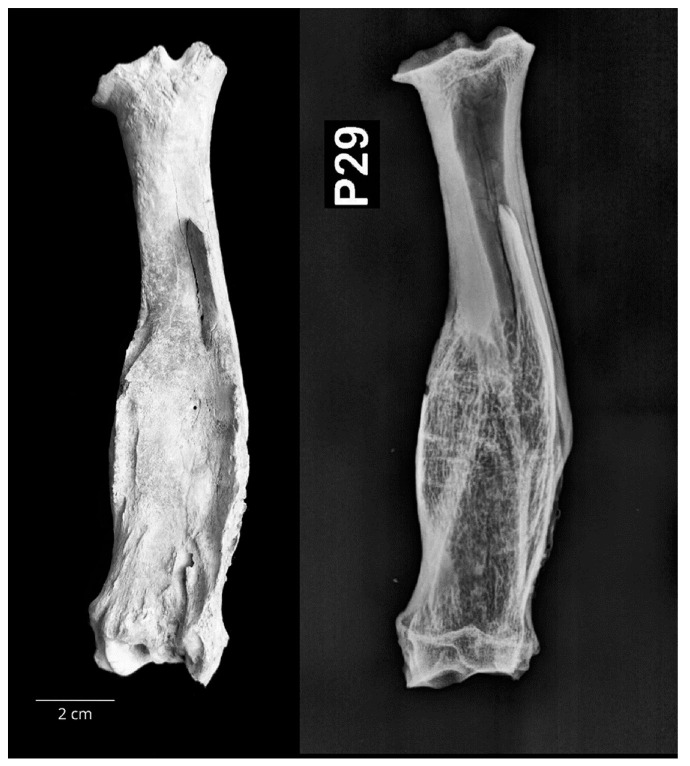
The right small ruminant radius (P29) with extensive bone swelling of the diaphysis. Radiological examination showed status post fracturam with the formation of the callus in the middle and the distal third of the diaphysis (**left**—caudal macromorphological view; **right**—sagittal radiological projection).

**Figure 10 vetsci-10-00361-f010:**
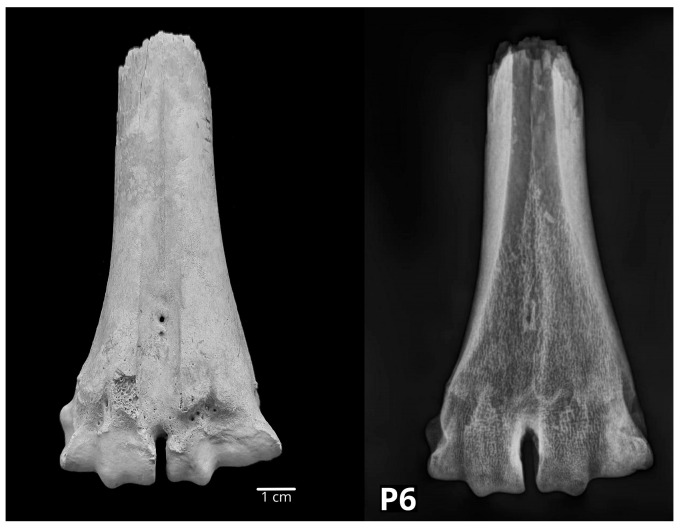
The distal half of a bovine metatarsal bone (P6) showing an unequal width of the articular surfaces of the head with the wider lateral articular surface confirmed radiologically as osteosclerosis (**left**—dorsal macromorphological view; **right**—sagittal radiological projection).

**Figure 11 vetsci-10-00361-f011:**
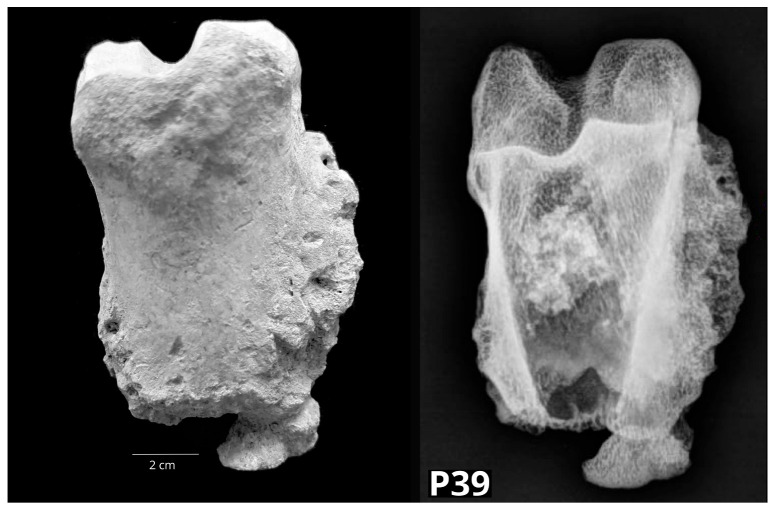
The bovine proximal phalanx (P39) with swellings of the compact bone tissue. Radiologically, osteolysis of the distal epiphysis and bone swelling is shown (**left**—dorsal macromorphological view; **right**—sagittal radiological projection).

**Figure 12 vetsci-10-00361-f012:**
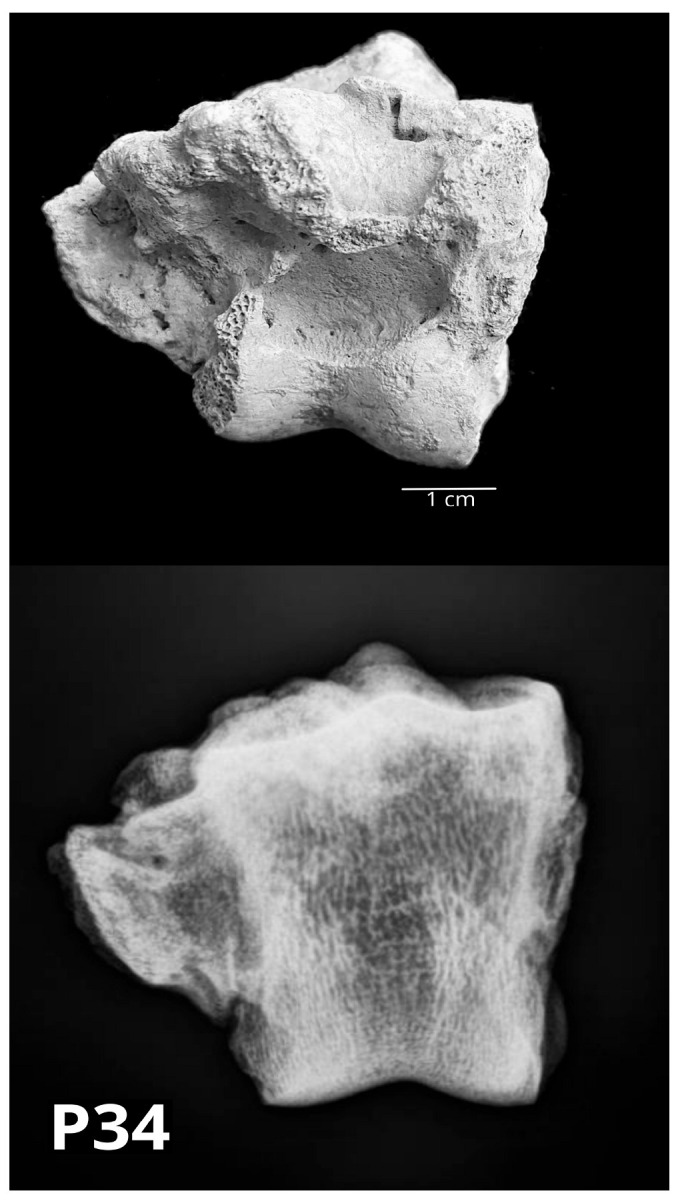
The bovine middle phalanx (P34) showed bone swellings at the whole medial side of the diaphysis. Radiological examination showed a suspected status post fracturam with periosteal bone swelling. Secondary arthritic changes and subchondral sclerosis were also found (**above**—palmar/plantar macromorphological view; **below**—sagittal radiological projection).

**Table 1 vetsci-10-00361-t001:** Bones with macrostructural changes according to their sample number, archaeological site, period of origin, skeletal position, side of the body, animal species and estimated age.

Sample No.	Site	Period	Bone Element	Side	Species	AGE
P1	PGV	High Middle Ages	MTC	left	cattle	>2–2.5 y
P2	PGV	High Middle Ages	talus	right	pig	
P3	PGV	High Middle Ages	MTC IV	right	pig	<2 y
P4	PGV	High Middle Ages	femur, dist	right	pig	<3.5 y
P5	PGV	Late Middle Ages	metapodium, dist		cattle	<2–2.5 y
P6	PGV	High Middle Ages	MTT, dist	left	cattle	<2–2.5 y
P7	PGV	Early Modern Period	mandibula, caput	left	cattle	
P8	PGV	Late Middle Ages	humerus, dist	right	small R	>11–13 m
P9	PGV	High Middle Ages	costa		small R	
P10	PGV	High Middle Ages	mandibula	left	cattle	
P11	PGV	High Middle Ages	vertebra lumbalis		pig	<4–7 y
P12	PGV	Early Modern Period	calcaneus	left	cattle	>3–3.5 y
P13	PGV	Early Modern Period	calcaneus	left	cattle	>3–3.5 y
P14	PGV	Early Modern Period	acetabulum	left	cattle	>7–10 m
P15	KRČ	High/Late Middle Ages	mandibula, rostral	right	small R	
P16	KRČ	High/Late Middle Ages	humerus, dist	right	pig	>1–1.5 y
P17	KRČ	High/Late Middle Ages	tibia, dist	left	pig	>2 y
P18	BIJELA	Early Modern Period	mandibula, rostral	left	goat	
P19	BAR	Late Middle Ages	MTC, prox	right	cattle	
P20	BAR	High/Late Middle Ages	ulna	right	small R	>2.5–3 y
P21	OCZP	Late phase of the Middle Bronze Age	mandibula	right	pig	>1.5–2 y
P22	OCZP	Early phase of the Middle Bronze Age	mandibula	left	pig	
P23	OCZP	Early phase of the Middle Bronze Age	vertebrae lumbales (4)		carnivore	
P24	GOR	Early Modern Period	radius, dist	left	horse	>3.5 y
P25	GOR	Early Modern Period	MTC, dist	right	cattle	>2–2.5 y
P26	GOR	Early Modern Period	phalanx proximalis		cattle	>1.5 y
P27	GOR	Early Modern Period	phalanx media		cattle	>1.5 y
P28	GOR	Early Modern Period	MTC, dist	right	cattle	>2–2.5 y
P29	GOR	Early Modern Period	radius	right	small R	>3–4 y
P30	ST NIC	Iron age	phalanx proximalis		cattle	>1.5 y
P31	ST NIC	Iron age	phalanx proximalis		cattle	>1.5 y
P32	ST NIC	Early Modern Period	phalanx proximalis		cattle	>1.5 y
P33	ST NIC	Early Modern Period	MTC, dist	right	cattle	>2–2.5 y
P34	NADIN	Antiquity	phalanx media		cattle	>1.5 y
P35	NADIN	Antiquity	os tarsi centroquartale	right	cattle	
P36	NADIN	Late Middle Ages/Early Modern Period	humerus	right	chicken	
P37	NADIN	Late Middle Ages/Early Modern Period	mandibula	left	small R	around 2.5–3 y
P38	NADIN	Antiquity	phalanx proximalis		cattle	>1.5 y
P39	NADIN	Antiquity	phalanx proximalis		cattle	>1.5 y
P40	NADIN	Antiquity	phalanx proximalis		cattle	>1.5 y
P41	NADIN	Antiquity and Late Antiquity	phalanx proximalis		cattle	>1.5 y
P42	NADIN	Antiquity and Late Antiquity	phalanx proximalis		cattle	>1.5 y
P43	NADIN	Antiquity and Late Antiquity	MTC, prox	right	cattle	
P44	NADIN	Early Iron Age	MTT, prox	left	small R	
P45	NADIN	Early Iron Age	MTC, prox	right	sheep	
P46	NADIN	Early Iron Age	MTC, prox	left	goat	
P47	NADIN	Early Iron Age	axis		small R	>4–5 y
P48	NADIN	Early Iron Age	os carpi radiale		cattle	
P49	NADIN	Early Iron Age	mandibula, rostral	left	cattle	
P50	NADIN	Iron Age	radius, prox	right	sheep	>3–3.5 y

Legend: PGV—Plemićki grad Vrbovec; KRČ—Krčingrad; BIJELA—Bijela; BAR—Barilović; OCZP—Osijek Ciglana-Zeleno Polje; GOR—Gorjani; ST NIC—St. Nicholas Church; NADIN—Nadin-Gradina; MTC—metacarpal; MTT—metatarsal; prox—proximal; dist—distal; small R—small ruminant; > older than; < younger than; y—years; m—months.

**Table 2 vetsci-10-00361-t002:** Taxonomic distribution of pathologically altered bones per archaeological site.

Site	Cattle	Small Ruminants	Pig	Horse	Carnivore	Chicken	Total
GORJANI	4	1		1			6
OCZP			2		1		3
BIJELA		1					1
PGV	8	2	4				14
BARILOVIĆ	1	1					2
KRČINGRAD		1	2				3
ST NICHOLAS	4						4
NADIN	10	6				1	17
Total	27	12	8	1	1	1	50
% of total	54%	24%	16%	2%	2%	2%	100%

**Table 3 vetsci-10-00361-t003:** List of all specimens according to etiological and radiological classification.

Aethiolog. Classif. *	ORAL CAVITY			Working/Keeping			TRAUMA	WITHOUT PATHOL. CHANGES
Radiol. Classif.	MET	INFL/ NEOPL	DEG	MET	INFL/NEOPL	Degenerative	TRAUMA	
CATTLE	mandible (P7)	mandible (P49)	mandible (P10)		centroquartal tarsal (P35)	metacarpal (P1)	metacarpal (P43)	calcaneus (P12)
					ph. prox (P39)	metapodium (P5)		calcaneus (P13)
					ph. prox (P41)	metatarsal (P6)		
					radial carpal (P48)	acetabulum (P14)		
						metacarpal (P19)		
						metacarpal (P25)		
						ph. prox. (P26)		
						ph. media (P27)		
						metacarpal (P28)		
						ph. prox (P30)		
						ph. prox (P31)		
						ph. prox (P32)		
						metacarpal (P33)		
						ph. media (P34)		
						ph. prox (P38)		
						ph. prox (P40)		
						ph. prox (P42)		
Total for cattle	1	1	1	0	4	17	1	2
SMALL RUM.		mandible (P18)		ulna (P20)		humerus (P8)	rib (P9)	
		mandible (P37)				metacarpal (P45)	mandible (P15)	
						metacarpal (P46)	radius (P29)	
						radius (P50)	metatarsal (P44)	
						axis (P47)		
Total for small rum.	0	2	0	1	0	5	4	0
PIG		mandible (P21) **				talus (P2)	metacarpal IV (P3)	mandible (P21) **
						lumbar vertebrae (P11)	femur (P4)	
						humerus (P16)	tibia (P17)	
							mandible (P22)	
Total for pigs	0	(+1)	0	0	0	3	4	1
HORSE						radius (P24)		
Total for horse	0	0	0	0	0	1	0	0
DOG						lumbar vertebrae (P23)		
Total for dog	0	0				1		
CHICKEN							humerus (P36)	
Total for chicken	0	0	0	0	0	0	1	0
Total	1	3 (+1)	1	1	4	27	10	3 (−1)
Total by group	5 (+1)			32			10	3 (−1)

Legend: * According to von den Driesch [42] and Bartosiewicz [2]. MET—metabolic; INFL/NEOPL—inflammatory/neoplastic; DEG—degenerative; ph. prox—phalanx proximalis. ** Specimen of pig mandible radiologically without pathological changes (−1), but macromorphologically with inflammatory/neoplastic alteration (+1).

## Data Availability

All data used in the current study are available from the corresponding author upon reasonable request.

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
