# Peer review of "Paleopathological Changes in Animal Bones from Croatian Archaeological Sites from Prehistory to New Modern Period"

_vetsci, 2023, doi:10.3390/vetsci10050361_

Round 1

Reviewer 1 Report

This is an easily readable, well-made article, that is interesting and useful mainly for zooarchaeologists.

A few minor suggestions.

The chronology of the periods used in the publication should be given, when the period is mentioned for the first time (e.g. lines 131-133),  since it vary in different regions.

Perhaps a chronology in the title would be useful?

Reviewer 2 Report

The paper approaches a topic that is not very frequent in archaeozoological studies. Authors stress nicely the causes of this situation, so I may consider this paper one of the few ones that approach pathological bone in archeozoology in a (little) broader perspective. Still, the very limited number of pathological specimens still is a major drawback for firm conclusions,  for an archaeozoologist (including myself) they may serve as a good reference point for further (correct) reporting of the finds. A strong poit of the approach is the use of RX and other imaging techniques that are usually available only for veterinarians.

I honestly do not find flaws or problems with the paper so I fully recommend the publishing of this in the present form

Reviewer 3 Report

The authors provide a comprehensive overview of the paleopathological findings from animal bones from Croatian sites. The overlap between veterinary and medical sciences is clear, this paper demonstrates yet another important interaction between different disciplines that utilise similar techniques. 

It would ideal to compare the findings in the current study to animals from these sites for comparison. That could be a separate study in itself. Nonetheless, comparing between archeological sites and animal bones obtained "today" would be of interest. 

Other techniques (micro CT, FTIR, XRD) would be exciting to explore to see how this type of data compares to animals as well as human skeletons (diet etc). 

The paper was interesting to read. The authors could provide additional detail in the introduction (paragraph 2) regarding the comment on diseas and bone structure. What disease(s) and what are the implications?

Much of the introduction reads like discussion, which is fine, just a comment on style.

Methods

Drying at room temperature would not be "dry". Does this matter? 

The authors could included what they are specifically lookin gfor in the radiographic scans rather than jusy saying they were analysed and described. What was analysed? Could a grading scale be used? 

The tables are very detailed which is great but could be formatted to be smaller? The macro images and X-rays are well presented. Thank you. 
